# Phylogeography of the second plague pandemic revealed through analysis of historical *Yersinia pestis* genomes

Maria A. Spyrou [1,2,19], Marcel Keller [1,3,19], Rezeda I. Tukhbatova[1,4], Christiana L. Scheib [5], Elizabeth A. Nelson[1,2], Aida Andrades Valtueña [1], Gunnar U. Neumann [1], Don Walker [6], Amelie Alterauge[7], Niamh Carty[6], Craig Cessford [8], Hermann Fetz[9], Michaël Gourvennec [10], Robert Hartle[6], Michael Henderson[6], Kristin von Heyking[3], Sarah A. Inskip [11], Sacha Kacki[12,13], Felix M. Key[14], Elizabeth L. Knox[6], Christian Later[15], Prishita Maheshwari-Aplin[8], Joris Peters[3,16], John E. Robb[8], Jürgen Schreiber[17], Toomas Kivisild [5,18], Dominique Castex[12], Sandra Lösch [7], Michaela Harbeck[3], Alexander Herbig [1], Kirsten I. Bos [1] & Johannes Krause [1,2]

The second plague pandemic, caused by *Yersinia pestis*, devastated Europe and the nearby regions between the 14[th] and 18[th] centuries AD. Here we analyse human remains from ten European archaeological sites spanning this period and reconstruct 34 ancient *Y. pestis* genomes. Our data support an initial entry of the bacterium through eastern Europe, the absence of genetic diversity during the Black Death, and low within-outbreak diversity thereafter. Analysis of post-Black Death genomes shows the diversification of a *Y. pestis* lineage into multiple genetically distinct clades that may have given rise to more than one disease reservoir in, or close to, Europe. In addition, we show the loss of a genomic region that includes virulence-related genes in strains associated with late stages of the pandemic. The deletion was also identified in genomes connected with the first plague pandemic (541–750 AD), suggesting a comparable evolutionary trajectory of *Y. pestis* during both events.

[1] Max Planck Institute for the Science of Human History, 07745 Jena, Germany. [2] Institute for Archaeological Sciences, University of Tübingen, 72070 Tübingen, Germany. [3] SNSB, State Collection for Anthropology and Palaeoanatomy Munich, 80333 Munich, Germany. [4] Laboratory of Structural Biology, Kazan Federal University, Kazan, Russian Federation 420008. [5] Institute of Genomics, University of Tartu, Riia 23b, 51010 Tartu, Estonia. [6] MOLA (Museum of London Archaeology), London N1 7ED, UK. [7] Department of Physical Anthropology, Institute for Forensic Medicine, University of Bern, 3007 Bern, Switzerland. [8] Department of Archaeology, University of Cambridge, Downing St, Cambridge CB2 3ER, UK. [9] Archaeological Service, State Archive Nidwalden, 6371 Nidwalden, Switzerland. [10] Archeodunum SAS, Agency Toulouse, 8 allée Michel de Montaigne, 31770 Colomiers, France. [11] McDonald Institute for Archaeological Research, University of Cambridge, Downing St, Cambridge CB2 3ER, UK. [12] PACEA, CNRS Institute, Université de Bordeaux, 33615 Pessac, France. [13] Department of Archaeology, Durham University, South Rd, Durham DH1 3LE, UK. [14] Institute for Medical Engineering and Sciences, Massachusetts Institute of Technology, Cambridge, MA 02139, USA. [15] Bavarian State Department of Monuments and Sites, 80539 Munich, Germany. [16] ArchaeoBioCenter and Department of Veterinary Sciences, Institute of Palaeoanatomy, Domestication Research and the History of Veterinary Medicine, Ludwig Maximilian University Munich, Kaulbachstr. 37/III, 80539 Munich, Germany. [17] Dig it! Company GbR, 86971 Peiting, Germany. [18] Department of Human Genetics, Katholieke Universiteit Leuven, 3000 Leuven, Belgium. [19]These authors contributed equally: Maria A. Spyrou, Marcel Keller. Correspondence and requests for materials should be addressed to M.A.S. (email: spyrou@shh.mpg.de) or to K.I.B. (email: bos@shh.mpg.de) or to J.K. (email: krause@shh.mpg.de)

One of the most devastating pandemics of human history was the second plague pandemic, which began with the infamous Black Death (BD, 1346–1353 AD) and continued with recurrent outbreaks in Europe, the Near East and North Africa until the 18th century AD[1,2]. Its causative agent, *Yersinia pestis*[3], is a highly virulent bacterium that causes bubonic, pneumonic, and septicaemic plague and today is maintained among wild rodent populations in eastern Europe, Asia, Africa and the Americas[4–6].

The first historically documented outbreaks of the second pandemic seem to have occurred in 1346 in the Lower Volga and Black Sea regions[1,7]. Subsequently, the bacterium dispersed through the rest of Europe over the next seven years, causing reductions in the human population estimated to be as high as 60%[1]. Recent studies on ancient *Y. pestis* DNA from medieval plague victims have contributed insights into these initial stages of the pandemic. Specifically, mid-14th-century *Y. pestis* genomes reconstructed from Saint-Laurent-de-la-Cabrerisse (southern France)[8], Barcelona (Spain)[9], London (England)[10] and Oslo (Norway)[8] were shown to be identical, suggesting the rapid dispersal of a single strain across Europe during the BD. Recently, the analysis of an additional low-coverage genome from Siena, Italy (BSS31)[8], revealed the purported existence of *Y. pestis* strain diversity during the BD, a possibility that should be further explored.

After the BD, plague was a common scourge in Europe as evidenced by the thousands of recorded epidemics it supposedly caused between 1353 and the late 18th century[2,11]. Whether these were caused by multiple introductions of the disease from an Asian source or by its local persistence in Europe is currently a topic of debate[9,12–14]. While data from climatic proxies are considered as supportive of the former hypothesis[12], genetic evidence is interpreted in both directions[8,9,13]. To date, ancient *Y. pestis* genomes from epidemics closely succeeding the BD in Europe have been sequenced from late-14th-century individuals in Bergen op Zoom (Netherlands), London (England) and the Middle Volga region of Russia. They cluster on a phylogenetic lineage that is a precursor to strains associated with the 19th-century third plague pandemic[9,15,16], and thus provide a link between medieval and modern plague. Moreover, *Y. pestis* genomes recovered from Ellwangen, Germany (1485–1627 calAD), and the Great Plague of Marseille in France (L'Observance, 1720–1722 AD) cluster on an independent lineage, here termed the "post-BD" lineage, that is to date unidentified among modern *Y. pestis* diversity. Both lineages descended from the strain associated with the BD and, hence, likely represent plague's legacy in or around Europe after 1353.

At present, the source of the second pandemic and the route that the bacterium followed during its course of entry into Europe remain hypothetical since genomic data from early outbreaks in western Russia have thus far been elusive. In addition, the limited number of published ancient *Y. pestis* genomes[9,10,14] challenges our ability to construct hypotheses regarding the number of lineages responsible for the numerous post-BD outbreaks in Europe[2,11] and whether they derived from a single or multiple disease reservoirs. Here, we take steps to overcome these limitations by expanding the number of available *Y. pestis* genomes from multiple time periods and locations in order to gain additional knowledge on the early stages of the second pandemic, and to study the genetic diversity of the bacterium present in Europe after the BD. Additionally, we present a reanalysis of recently published data from the same time period[8]. Our results support the entrance of *Y. pestis* into Europe through the east during the initial wave of the pandemic and consistently demonstrate an absence of genetic diversity in the bacterium during the BD. Moreover, our genomic analysis of post-BD outbreaks from

central and western Europe suggests the local diversification of an extinct *Y. pestis* lineage between the late-14th and 18th centuries that may have resided in more than one disease reservoir.

## Results

**Sample screening for signatures of *Y. pestis* DNA.** Two approaches were used for the assessment of *Y. pestis* DNA in tooth specimens ($n = 206$) from ten archaeological sites spanning the 14th–17th centuries AD in Europe (Fig. 1, Supplementary Figs. 1–10 and Supplementary Note 1). First, a qPCR screening approach was employed for detection of the *Y. pestis*-specific gene, *pla*, located on the pPCP1 plasmid[17] in 180 specimens from the cities of London ($n = 40$) in England, Toulouse ($n = 42$) in France, Brandenburg an der Havel[13] ($n = 3$), Landsberg am Lech ($n = 10$), Manching-Pichl[13] ($n = 28$), Nabburg ($n = 12$) and Starnberg ($n = 3$) in Germany, Laishevo ($n = 10$) in the Volga region of Russia, and Stans ($n = 32$) in Switzerland. Extracts from 41 teeth across these sites tested positive for *pla* (Supplementary Table 1). All extraction negative controls were free of amplification products. Amplification products from putatively positive individuals were not sequenced, as the presence of *Y. pestis* was subsequently assessed through whole-genome capture and high-throughput Illumina sequencing.

In addition, shotgun Next Generation Sequencing (NGS) data from individuals unearthed at the New Museums site (Augustinian Friary) in Cambridge ($n = 26$) were screened for *Y. pestis* with the MEGAN alignment tool (MALT)[18] (see Methods). The output was post-processed within the pathogen screening pipeline HOPS[19]. The assessment of shotgun NGS reads produced from non-uracil-DNA-glycosylase (non-UDG) libraries revealed the potential presence of *Y. pestis* DNA in four individuals (Supplementary Table 2, Supplementary Fig. 11).

***Y. pestis* in-solution capture and whole-genome reconstruction.** We prepared UDG-treated libraries[20,21] from all putatively positive samples and used a *Y. pestis* whole-genome in-solution capture approach[22] combined with high-throughput sequencing for the retrieval of 1,299,105–79,055,317 raw reads per sequenced library. All data were mapped against the *Y. pestis* CO92 reference genome (NC_003143.1)[3]. This resulted in 86,278–3,822,030 unique mapping reads yielding 1.1–80.1-fold coverage across 34 individuals that span the time transect between the 14th and 17th centuries in Europe (Supplementary Table 3). More specifically, we could retrieve two *Y. pestis* genomes from Cambridge (England), five from London (England), one from Toulouse (France), three from Nabburg, two from Manching-Pichl[13], one from Starnberg, one from Landsberg am Lech, two from Brandenburg an der Havel[13] (all from Germany), two from Laishevo (Russia) and 15 from Stans (Switzerland). Of those, 24 isolates showed at least 50% of the reference genome covered at 5-fold (Table 1), which allowed for their confident inclusion in phylogenetic analysis. In addition, we nearly tripled the genomic coverage of the published "549_O" isolate from Ellwangen, Germany (now reaching 14.1-fold), which was previously processed by array-based capture using a different probe design[9] (Supplementary Table 3).

***Y. pestis* phylogenetic reconstruction.** To infer genetic relationships between the new and previously published *Y. pestis* isolates, we constructed phylogenies using the maximum likelihood (ML) method, allowing for up to 3% missing data (97% partial deletion) to accommodate lower coverage genomes. As a reference dataset, we used 233 modern isolates[23–27] (as listed in ref. [28]), which represent most of the published *Y. pestis* genetic diversity. In addition, we included previously published second

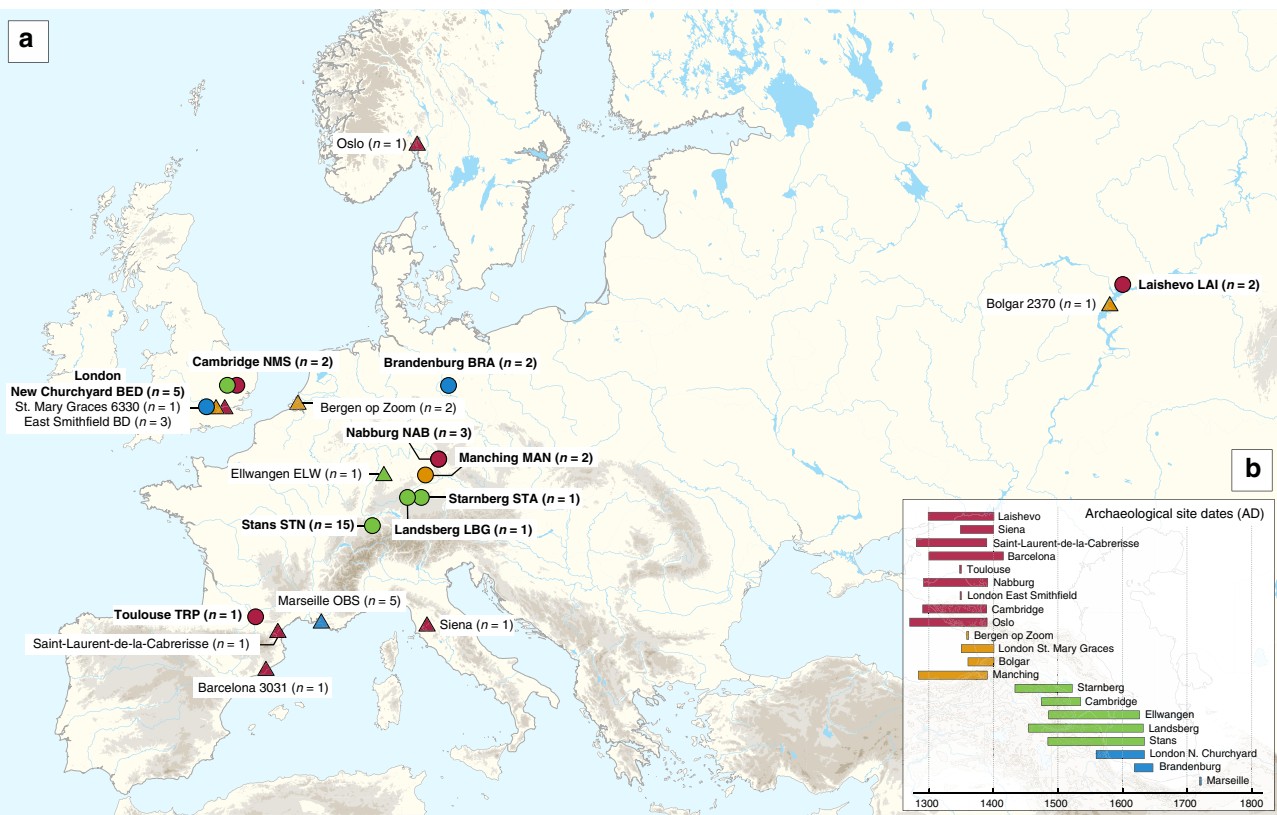

**Fig. 1** Archaeological site locations and chronologies. **a** Map showing the geographic locations of archaeological sites from which second pandemic (14th- to 18th-century AD) *Y. pestis* genomes have been reconstructed (≥1-fold). The number (*n*) of genomes obtained from each site is shown in brackets. Locations of previously published genomes appear in triangles, whereas genomes that are newly described in this study appear in circles (labels in bold). Base map purchased from [vectormaps.de]. **b** Specimen chronologies combining archaeological and radiocarbon dates of previously published and new second plague pandemic isolates (see Supplementary Note 1 and Supplementary Table 3)

pandemic isolates (*n* = 15)[8–10,14], a 6th-century AD isolate from Germany[29], a 2nd- to 3rd-century AD isolate from the Tian Shan mountains in Kyrgyzstan[30], as well as three Bronze Age isolates from the Altai and Volga regions[31,32] (Supplementary Fig. 12).

All newly reconstructed genomes appear on Branch 1 and are closely related to the previously published second pandemic isolates from Europe (Fig. 2), thus confirming their authenticity. In addition, they seem to represent a diverse group of strains that were present across Europe between the 14th and 18th century AD (Fig. 2, Supplementary Data 1). A number of genomes (NAB005, BRA003, STN011 and STN004) were excluded from further analyses as they showed evidence of excess heterozygosity, which is atypical of bacterial genomes (Supplementary Fig. 13). This phenomenon likely arises from enrichment of non-target DNA stemming from closely related organisms, an issue frequently encountered in ancient metagenomic datasets[18,29,33]. Moreover, these genomes had notably longer branch lengths in comparison to other contemporaneous isolates from the same archaeological contexts (Supplementary Fig. 14). Their assessment using the recently developed SNPEvaluation tool[28] (see Methods) classified their private SNP calls as false-positive, suggesting that the observed branch lengths are erroneous (Supplementary Data 2). Similarly, the previously published SLC1006 and BSS31 genomes[8] were also excluded from further analyses as they also showed high heterozygosity (Supplementary Fig. 15) and exceedingly longer branch lengths compared to other 14th-century *Y. pestis* genomes (Supplementary Figs. 14 and 16).

Our phylogenetic reconstruction shows that the LAI009 isolate from Laishevo is ancestral to the BD isolates from southern, central, western and northern Europe, as well as to the previously published late 14th-century isolates from London (6330)[10] and Bolgar City[9] (Fig. 2). This genome possesses only one derived SNP distinguishing it from the N07 polytomy that gave rise to Branches 1–4 (Fig. 2; Supplementary Data 1)[23]. Since all other second pandemic genomes share an additional derived SNP on Branch 1, we interpret LAI009 as the most ancestral form of the strain that entered Europe during the initial wave of the second pandemic that has been identified to date. Regarding the central and western European genomes, NAB003 from Nabburg does not show differences compared to previously published BD genomes from London and Barcelona[9,10]. In addition, NMS003 from Cambridge was genotyped based on inspection of its SNP profile, despite it not fulfilling the genomic coverage criteria for inclusion in our phylogenetic analysis (Supplementary Table 3), as its archaeological context makes it distinct from other *Y. pestis*-positive individuals from the same site (see Supplementary Note 1). As a result, SNP inspection classified it as potentially identical to other BD genomes (Supplementary Data 3). By contrast, certain isolates associated with the BD period are seemingly distinct. For example, TRP002 from Toulouse, which dates to 1347–1350 based on archaeological evidence, forms its own unique branch (Fig. 2; Supplementary Data 1). Qualitative assessment of eight unique SNPs in TRP002 with SNPEvaluation[28] classified them as potential false-positives (see Methods, Supplementary Data 2). In addition, after visual inspection, all such variants appear in regions of the genome where reads from diverse sources seem to be mapping (Supplementary Fig. 17) and, therefore, were considered to be of exogenous origin. Similarly, we assessed one unique SNP identified in our re-analysis of the recently published OSL-1 genome from Oslo, Norway[8] (Fig. 2).

**Table 1 Post-capture sequencing statistics of all new *Yersinia pestis* genomes that passed quality criteria for inclusion in phylogenetic analysis**

| Sample name | Site name | Date (AD) | Uniquely mapping reads | Endogenous DNA post enrichment (%) | Mean fold coverage | Genome covered ≥ 5-fold (%) | Average fragment length (bp) | GC content (%) |
|---|---|---|---|---|---|---|---|---|
| BED030.A0102 | New Churchyard, London | 1560-1635[a] | 3,624,482 | 36.2 | 80.1 | 93.6 | 102.9 | 48.5 |
| BED028.A0102 | New Churchyard, London | 1560-1635[a] | 2,665,238 | 22.2 | 37.2 | 91.4 | 65.0 | 49.0 |
| BED034.A0102 | New Churchyard, London | 1560-1635[a] | 1,371,698 | 10.5 | 18.3 | 89.1 | 62.2 | 49.2 |
| BED024.A0102 | New Churchyard, London | 1560-1635[a] | 1,000,524 | 18.1 | 12.6 | 84.7 | 58.5 | 49.1 |
| BRA001.A0101 | Domlinden 12, Brandenburg an der Havel | 1618-1648[b] | 2,387,557 | 23.2 | 23.8 | 92.0 | 46.4 | 47.5 |
| LAI009.A0101 | Laishevo III, Laishevo | 1300-1400[b] | 2,549,926 | 23.9 | 28.4 | 92.1 | 51.8 | 48.0 |
| LBG002.A0101 | Kirchhof St. Johannis, Landsberg | 1455-1632[a] | 621,713 | 27.9 | 7.2 | 66.4 | 54.2 | 49.9 |
| MAN008.B0101 | St. Leonhardi, Manching-Pichl | 1283-1390[a] | 1,974,399 | 44.9 | 25.8 | 88.7 | 60.8 | 50.8 |
| NAB003.B0101 | Sankt Johans Freidhof, Nabburg | 1292-1392[a] | 684,029 | 33.5 | 8.1 | 70.2 | 54.8 | 49.7 |
| NMS002.A0101 | Augustinian Friary (New Museums site), Cambridge | 1475-1536[a,b] | 855,185 | 2.3 | 12.5 | 94.8 | 67.9 | 47.3 |
| STA001.A0101 | Possenhofener Str. 3, Starnberg | 1433-1523[a,b] | 1,110,049 | 9.9 | 11.7 | 84.3 | 49.0 | 44.7 |
| STN014.A0101 | Nägeligasse, Stans | 1485-1635[a] | 3,822,030 | 48.2 | 55.3 | 93.0 | 67.3 | 48.9 |
| STN020.A0101 | Nägeligasse, Stans | 1485-1635[a] | 2,020,769 | 44.3 | 28.2 | 90.3 | 64.8 | 48.5 |
| STN021.A0101 | Nägeligasse, Stans | 1485-1635[a] | 1,588,442 | 35.1 | 21.7 | 88.6 | 63.7 | 48.5 |
| STN019.A0101 | Nägeligasse, Stans | 1485-1635[a] | 1,325,076 | 35.3 | 18.7 | 87.1 | 65.8 | 49.1 |
| STN007.A0101 | Nägeligasse, Stans | 1485-1635[a] | 1,293,507 | 32.8 | 18.0 | 86.7 | 64.8 | 49.4 |
| STN002.A0101 | Nägeligasse, Stans | 1485-1635[a] | 935,795 | 27.6 | 12.7 | 83.3 | 63.0 | 48.4 |
| STN008.A0101 | Nägeligasse, Stans | 1485-1635[a] | 875,153 | 30.2 | 11.7 | 77.7 | 62.5 | 50.1 |
| STN013.A0101 | Nägeligasse, Stans | 1485-1635[a] | 714,482 | 24.5 | 9.2 | 73.8 | 59.9 | 49.0 |
| TRP002.A0101 | Trente-Six Ponts 16, Toulouse | 1347-1350[b] | 632,303 | 19.8 | 5.9 | 50.9 | 43.2 | 48.7 |

[a]Dates based on radiocarbon dating of collagen
[b]Dates based on archaeological context information

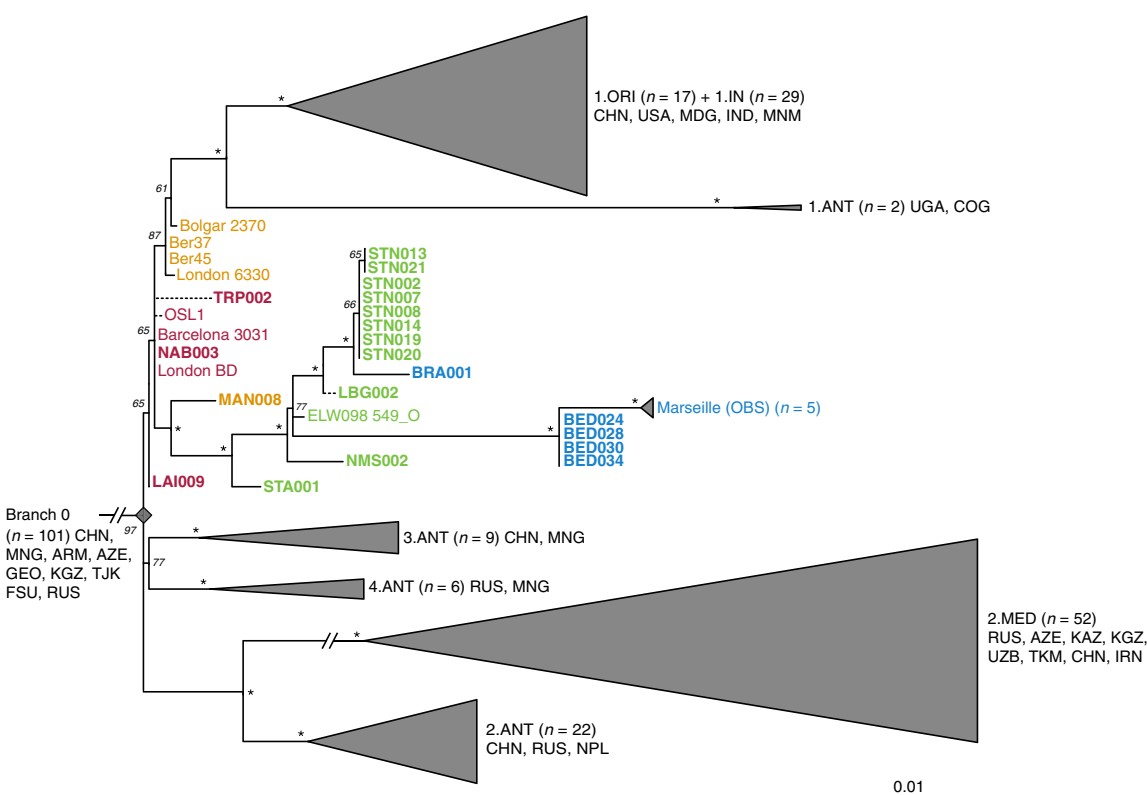

**Fig. 2** Phylogenetic positioning of second pandemic strains. A maximum likelihood phylogeny was generated allowing for up to 3% missing data (97% partial deletion) and considering a total of 6,058 single nucleotide polymorphisms (SNPs). The image shows a graphical representation of Branches 1–4 (see Supplementary Fig. 12 for a complete phylogeny), to emphasise the phylogenetic positioning of the new and previously published second pandemic strains (labels of new 14th- to 17th-century strains appear in bold). Dashed branches denote uncertainty in the private SNP calls of the respective genomes. Sub-clades of published genomes are collapsed to enhance tree visibility. Numbers ($n$) in brackets indicate the number of strains represented in each collapsed branch. Node support was estimated using 1,000 bootstrap iterations. Nodes that have bootstrap values of ≥95 are indicated by asterisks (*). Scale denotes substitutions per site. Geographic abbreviations of modern strain isolation locations are as follows: China (CHN), United States of America (USA), Madagascar (MDG), India (IND), Myanmar (MNM), Congo (COG), Uganda (UGA), Mongolia (MNG), Nepal (NPL), Iran (IRN), Kazakhstan (KAZ), Kyrgyzstan (KGZ), Tajikistan (TJK), Armenia (ARM), Georgia (GEO), Azerbaijan (AZE), Uzbekistan (UZB), Turkmenistan (TKM), Russia (RUS) and unspecified regions of the Former Soviet Union (FSU)

Visual inspection revealed it as a low-quality C-to-T transition that could be confined by aDNA damage (Supplementary Fig. 18). Finally, despite exclusion of BSS31 (Siena, Italy) from phylogenetic analysis, two previously identified unique SNPs in this genome were manually inspected, since they were presented as evidence for *Y. pestis* genetic diversity in Europe during the BD[8]. Importantly, BLASTn analysis of reads overlapping those regions (Supplementary Fig. 18, Supplementary Data 4 and 5) showed a 100% identity to environmental or other enteric bacterial species, but not to *Y. pestis*. We, hence, conclude that apart from LAI009 all reconstructed genomes associated with the initial pandemic wave have identical genotypes. In addition, we note that structural rearrangements could provide alternative means of genetic diversity. Although architectural differences are vastly abundant among modern *Y. pestis* genomes[34], their assessment in ancient *Y. pestis* is limited by the short read aDNA data produced here.

We find a number of genomes grouping with the previously described "post-BD" lineage together with published strains from Ellwangen (ELW098/549_O), Germany (1486–1630)[9], and Marseille, France (1720–1722)[14], which are descended from the European BD isolates (Fig. 2; Supplementary Data 1). Here, we identify the earliest evidence of this lineage in a 14th-century isolate from Manching-Pichl (MAN)[13] (see Supplementary

Note 1), which is followed by the more derived 15th- to 17th-century isolates from Starnberg (STA), Landsberg (LBG), Stans (STN) and Cambridge (NMS), as well as the 17th-century Brandenburg an der Havel (BRA)[13] and London (BED), all of which provide further evidence for plague's continuous presence in Europe after the BD. Of note, we retrieved eight nearly identical genomes from Stans (STN, maximum one SNP difference in two of eight genomes; mean SNP distance $d = 0$), and together with the four identical genomes from 17th-century London (BED) ($d = 0$), the five previously published nearly identical genomes from Marseille (OBS, maximum one SNP difference in one of five genomes, $d = 0$), and the seven identical BD isolates from various regions in Europe ($d = 0$), our results demonstrate low genetic diversity of the bacterium within local outbreaks and/or major epidemics of the second pandemic. In addition, we find that this "post-BD lineage" gave rise to (at least) two distinct clades within Europe, with the Ellwangen isolate being positioned closest to an apparent population split (Fig. 2). From this divergence, one clade gave rise to the strains associated with outbreaks in Germany and Switzerland (15th–17th century AD), and the second encompassed strains from 17th-century London (BED) and 18th-century Marseille (OBS). Notably, these two clades show dissimilar rates of substitution accumulation. For example, the mean SNP distance between the Ellwangen genome

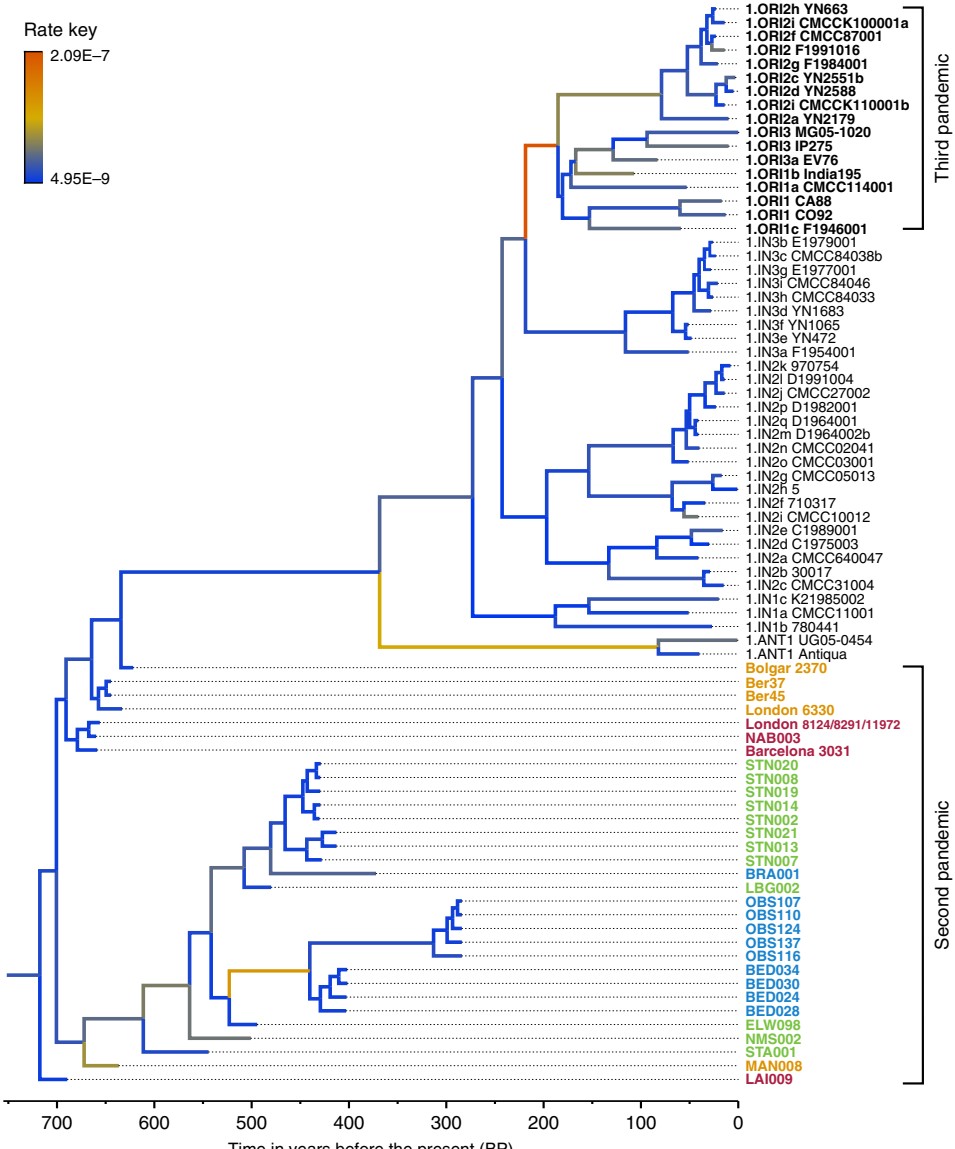

**Fig. 3** Substitution rate variation across the *Y. pestis* Branch 1. The figure presents a maximum clade credibility (MCC) phylogenetic tree generated using BEAST v1.8[85] (rooted with 2.MED KIM10—outgroup not shown). The tree was viewed in FigTree v1.4 (http://tree.bio.ed.ac.uk/software/figtree/), and modified so that branch colours represent mean substitution rates (substitutions per site per year). The tree depicts the substitution rate variation across Branch 1 of the *Y. pestis* phylogeny, which ranges from 2.09E-7 (highest-red) to 4.95E-9 (lowest-blue) substitutions per site per year (see rate key). The isolates used for this analysis overlap with the ones used for the SNP and maximum likelihood phylogenetic analysis (see Supplementary Fig. 12), with the exception of the TRP002 and OSL1 genomes since their private SNP calls are likely affected by environmental contamination and other representative genomes exist in our dataset from the BD time period (1346–1353 AD). Labels of genomes associated with the second and third plague pandemics appear in bold. The mean substitution rate across the tree (including 2.MED KIM10) was calculated to 2.85E-8 substitutions per site per year. Lengths of branches are scaled to represent sample ages, and the depicted Branch 1 sequences are estimated to represent 731 years (95% HPD: 672–823) of *Y. pestis* evolution. The time scale is shown in years before the present (BP), where present denotes the most recently isolated modern *Y. pestis* strain (year 2005)

(ELW098/549_O) and the London (BED) genomes ($d = 45$) is double that observed between Ellwangen and Brandenburg (BRA, $d = 22$), despite an assumption of them being contemporaneous (early 17th century AD) based on archaeological dating (Fig. 2; Supplementary Table 1; Supplementary Note 1).

**Analysis of substitution rate variation in *Y. pestis*.** We used the Bayesian framework BEAST v1.8 in order to make an assessment of substitution rate variations across the genealogy of Branch 1 ($n = 80$), retaining high-quality second pandemic *Y. pestis* genomes and using available calibration points in our modern and

ancient datasets (Supplementary Data 6). Previous studies have demonstrated that overdispersion among *Y. pestis* branch lengths is unlikely a result of natural selection, and have rather suggested a link between rate acceleration and geographic expansion of certain lineages during epidemic spread[16,23]. Our analysis based on the coalescent skyline model (Fig. 3, Supplementary Fig. 19) suggests an over 40-fold difference between the fastest and slowest substitution rates identified on Branch 1 (Fig. 3). In particular, we observe the fastest rates in three internal branches (Fig. 3). The first spans the genetic distance between the strains from Ellwangen (549_O) and London (BED), and supports the conflicting branch lengths of BED and BRA strains described

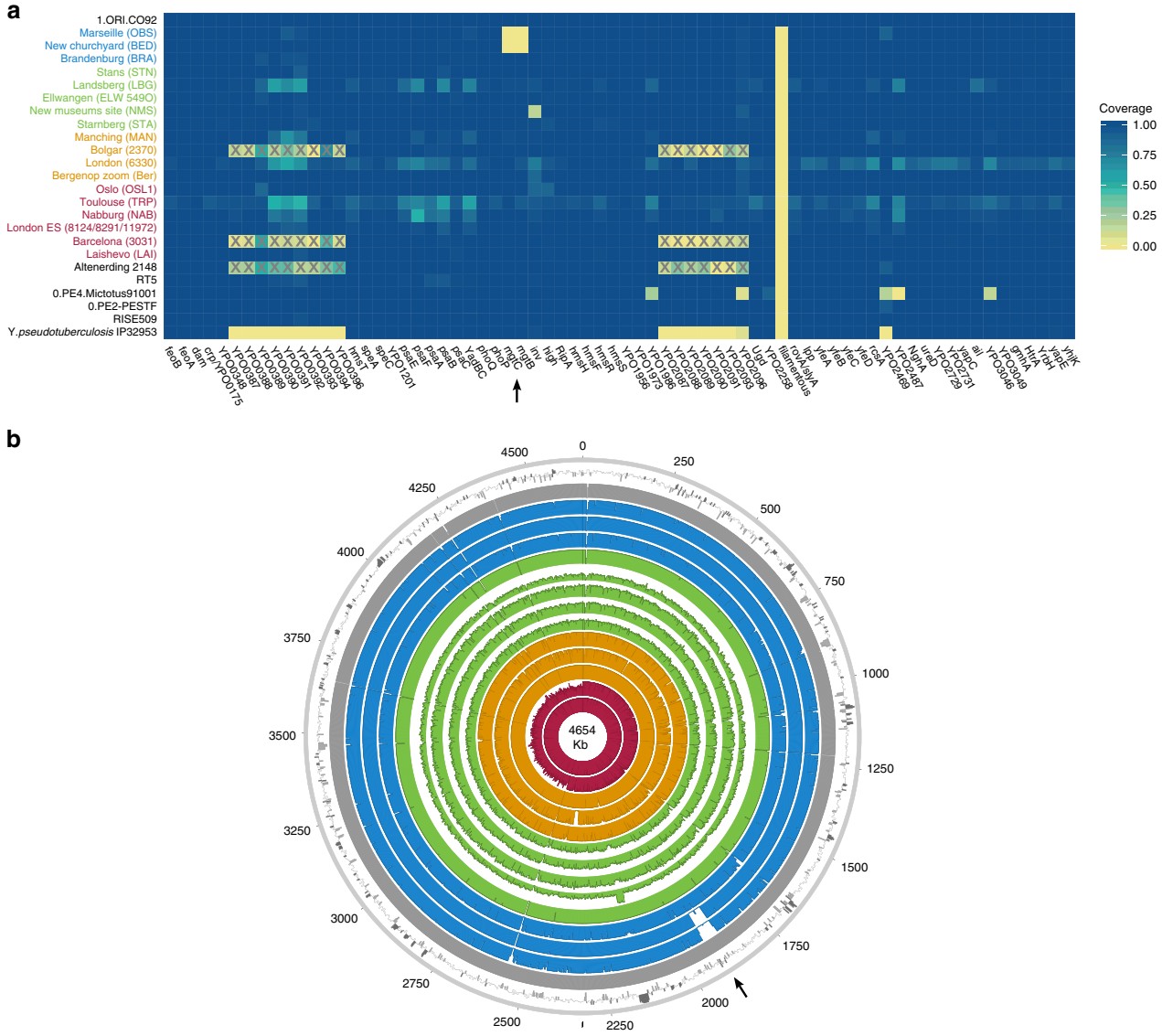

**Fig. 4** Assessment of chromosomal and gene-specific coverage in *Y. pestis*. **a** A comparison of genetic profiles was performed across newly reconstructed and previously published second pandemic genomes (in red, orange, green and blue). Here, we show an assessment of the presence or absence of 80 previously defined[36] potential virulence and evolutionary determinants across the *Y. pestis* chromosome. Published genomes from the Bronze Age period[31,32] (RISE509 and RT5), from the first pandemic[29] (6th-century Altenerding 2148), from modern-day isolates (0.PE2, 0.PE4 and 1.ORI)[23], as well as *Y. pseudotuberculosis* IP32953[60], are also shown for comparative purposes. The colour scale ranges from 0 (not covered—yellow) to 1 (entirely covered—blue) according to the relative proportion of gene/locus covered. The heatmap was plotted in R version 3.4.1[82] using the ggplot2 package[89]. Boxes marked with "X" indicate genomic loci that were not part of the *Y. pestis* probe design when the respective isolates were captured[9,29]. Refer to Supplementary Fig. 20 for presence/absence of virulence-associated genes across the pMT1, pPCP1 and pCD1 plasmids. **b** Chromosomal coverage plots made with the Circos[90] software. The plots were constructed to a maximum coverage of 20-fold, and the average coverage was calculated over 3,000-bp windows. Genomes are shown in chronological order from oldest (innermost circle) to youngest (outermost circle) as follows: LAI009, London BD 8124/8291/11972 (BD representative), Ber45, Bolgar 2370, MAN008, STA001, NMS002, ELW098/549_O, LBG002, STN014, BRA002, BED030, OBS137 and the reference genome CO92. The outermost ring represents fluctuations in GC content (%) across CO92, where dark and light grey bars show deviations from the genomic mean (47.6%) by at least one standard deviation

earlier (Fig. 3 and Supplementary Data 7). The second is the branch leading to the 1.ANT strains isolated from Africa (Congo and Uganda) (Fig. 3 and Supplementary Data 8). The broad history of 1.ANT and the time period associated with its establishment in Africa are unknown, though an introduction from Eurasia has been hypothesised[9,35]. The third, which displays the fastest rate within the entire Branch 1, is the branch leading to 1. ORI isolates (Fig. 3 and Supplementary Data 9), which is associated with the global spread of *Y. pestis* via maritime routes during the third plague pandemic (1894–1950s)[15,16]. Our results, therefore, support the idea of faster substitution rates during

epidemic spread, here particularly noticeable for lineages known to have expanded over wide geographic areas.

**Analysis of virulence-associated genomic profiles**. To investigate the genomic profiles of all newly reconstructed genomes, we analysed the presence or absence of potential virulence-associated and evolutionary determinant genes located on the *Y. pestis* chromosome (Fig. 4a) and plasmids (Supplementary Fig. 20)[36,37], in comparison to published representatives of ancient and modern strains. We find that the genetic profiles of some of the

previously characterised historical strains are influenced by the capture design used for their retrieval. Specifically, the second pandemic genomes "Bolgar 2370" and "Barcelona 3031" (ref. [9]) and the first pandemic genome "Altenerding 2148" (ref. [29]) seem to lack coverage in certain *Y. pestis*-specific regions, since *Yersinia pseudotuberculosis* was previously used as a probe-design template for their enrichment[9,29] (Fig. 4a). Regarding the newly reconstructed strains, we find that most possess all analysed genes with the exception of the New Churchyard (BED) and Marseille (OBS) strains that lack the magnesium transporter genes *mgtB* and *mgtC*, as well as the Cambridge (NMS002) strain that is lacking the *inv* gene (Fig. 4). While invasin is associated with epithelial colonisation of *Y. pseudotuberculosis* and *Yersinia enterocolitica*, it is known to have been inactivated in *Y. pestis*[38]. By contrast, magnesium transporters are considered vital for *Y. pestis* intracellular survival under low $Mg^{2+}$ conditions[39], such as those encountered within macrophage phagosomes. Specifically for *Y. pestis*, *mgtB* disruption has been associated with a decreased ability for macrophage invasion resulting in its attenuated virulence in mice[40]. Both *mgtB* and *mgtC* are present in all 233 modern *Y. pestis* genomes used in our comparative dataset. We explored these gene deletions in greater detail using BWA-MEM and identified them as part of a 49-kb missing region within the BED and OBS genomes (1,879,467–1,928,869 on CO92) (Fig. 4b, Supplementary Fig. 21) flanked by an *IS100* element immediately following its downstream end, which is consistent with previously characterised disruptions or losses of *Y. pestis* genomic regions via insertion elements[41]. Apart from *mgtB* and *mgtC*, this region encompasses a set of 34 additional genes that code for both characterised and hypothetical proteins, most of which seem to be associated with phenotypic characteristics that appear inactivated in *Y. pestis* such as motility and chemotaxis as well as few genes associated with metabolism, structure synthesis and environmental stress response (Supplementary Fig. 21, Supplementary Table 4). In addition, the clade encompassing this deletion is associated with some of the late outbreaks of the second plague pandemic, i.e. during the 17th century in London, England (BED) (see Supplementary Note 1), and during the 18th-century Plague of Marseille, in France (OBS 1720–1722 AD)[14], which was one of the last major epidemics that occurred in continental Europe[42]. Intriguingly, a nearly identical genomic deletion (45 kb), also including the *mgtB* and *mgtC* virulence-associated genes, was recently identified in ancient isolates from France (LVC, LSD)[28] sequenced from victims of the first plague pandemic (6th–8th centuries AD)[28]. These genomes are described elsewhere and date within a wide temporal interval (550–650 AD), though based on existing data they appear to be the youngest first pandemic isolates sequenced to date[28].

## Discussion

A series of studies have sufficiently demonstrated the preservation of *Y. pestis* in ancient human remains from a wide temporal transect[8–10,14,22,29,31,32,43]. This study presents an extensive sampling of multiple European epidemic burials from the period between the 14th and 17th centuries in order to gain a more complete picture of *Y. pestis*' genetic history during the second plague pandemic. Here, we nearly triple the amount of genomic data available from that time period (Fig. 1, Table 1 and Supplementary Table 3) and integration with existing datasets reveals key aspects regarding the initiation and progression of the second plague pandemic in Europe.

Based on historical sources alone, it has been difficult to determine the time at which *Y. pestis* first reached different parts of western Russia[7]. A commonly accepted view dates its arrival in the southwest, particularly in cities of Astrakhan and Sarai, in 1346[1,44] with subsequent spread into southern Europe from the Crimean peninsula. On the other hand, the dispersal of plague into northwestern Russia (i.e. in the cities of Pskov and Novgorod[7,44]) may have followed an alternative route via the Baltic Sea, occurring at the end of the BD between 1351 and 1353[1,7,44]. Such a notion of plague's expansion from northern Europe eastwards is also supported by published ancient genomic data from the late 14th-century Middle Volga region of Russia[9], though other scenarios may come to light with incorporation of additional genomic and historical data. Importantly, through analysis of our new strain from Laishevo (LAI009), which is phylogenetically ancestral to all second pandemic strains sequenced to date (Fig. 2), we provide evidence for the bacterium's presence in the same region, ~2000 km northeast of the Crimean peninsula, prior to reaching southern Europe in 1347–1348[1] (currently represented by strains from Siena, Saint-Laurent-de-la-Cabrerisse, Barcelona and Toulouse[8,9]). These results suggest that the N07-derived SNP previously termed "p1"[9] (Fig. 2, Supplementary Fig. 12), that is common to all other second pandemic strains, was likely acquired within Europe during the onset of the BD. In addition, given the proximity of the LAI009 genome to the N07 node often associated with the initiation of the BD (Fig. 2, Supplementary Fig. 12)[23], further data will be necessary to accurately re-evaluate the geographic origin of Branch 1. Previous analyses have proposed East Asia as the mostly likely candidate for the N07 polytomy[10,23] (Fig. 2). Such claims, however, cannot yet be verified given; (1) the apparent East Asian sampling bias of modern isolates[23,45], (2) the lack of molecular evidence from East Asia dating to the early 14th century and (3) the scarcity of historical documentary sources from this region describing precise disease symptoms[46]. In addition, recently published modern *Y. pestis* genomes from Central Asia show a rich diversity in the local plague foci[26,27], and further sampling from these regions has the potential to inform hypotheses on plague movement and evolution.

The identification of low genomic diversity during the initial wave of the second pandemic becomes particularly informative when attempting to reconstruct the spread of plague after 1353. Previous research based on climatic proxies[12] as well as PCR[47] and genomic[8] data have proposed multiple introductory waves of *Y. pestis* into Europe as the main source for the post-BD outbreaks recorded until the 18th century. Here, using previously published[8–10,14] and new whole-genome data from 20 archaeological sites, we identify that all genomes associated with post-BD outbreaks in Europe derived from a single ancestral strain that was present in southern, central, western and northern Europe during the BD. We, therefore, interpret the current data as supporting a single entry of *Y. pestis* during the BD, though additional interpretations may arise through the discovery of unsampled diversity in western Eurasia. Subsequent to its entry, we observe the formation of two sister lineages (Fig. 2). The first lineage is responsible for the bacterium's possible eastward expansion after the BD. It contains strains from late-14th-century Bergen op Zoom, London (6330)[10] and the city of Bolgar (2370)[9], as well as extant strains from Africa (1.ANT)[48], and, most importantly, a worldwide set of isolates associated with the third pandemic (1.ORI, 19th–20th centuries)[15,16,23] (Fig. 2). The second, here termed the "post-BD lineage", is characterised by a profound genomic diversity identified within Europe that seems to have been restricted to the second pandemic, as no modern descendants have been identified for this lineage to date. It is represented by historical genomes isolated from 14th- to 18th-century Germany (MAN, STA, ELW, LBG and BRA), Switzerland (STN), England (NMS, BED) and France (OBS) (Fig. 2), suggesting that it persisted in Europe or its vicinity and caused infections over a wide geographic range. The fact that this lineage

has no identified modern descendants is likely related to the disappearance of plague from Europe in the 18th century, possibly due to extinction of local reservoirs, as previously suggested[9].

We find that the "post-BD lineage" gave rise to (at least) two distinct clades that separate the strains identified in Central Europe during the 15th–17th centuries, and those identified in 17th- to 18th-century England and France. Their distinction is corroborated not only by their genetic and geographic separation (Fig. 2), but also by potential differences in their genomic profiles (Fig. 4) and substitution rates (Fig. 3). The clade that exhibits a slower substitution rate is mainly represented by temporally and genetically closely related isolates from Germany and Switzerland (Fig. 2), which could indicate endemic circulation of the bacterium in that region. Such an observation may be compatible with the hypothesis of an Alpine rodent reservoir facilitating the spread of plague in Central Europe after the BD[49], although a possible sampling bias should be noted since the majority of our data derive from this region. On the other hand, the clade that exhibits a faster substitution rate (Fig. 3) appears to have had a wider geographic distribution. Given that both Marseille and London were among the main maritime trade centres in Europe during that time, it is plausible that introduction of the disease in these areas occurred via ships[50], although sources favouring local epidemic eruptions also exist[51]. Previous studies have demonstrated that transmission of *Y. pestis* via steamships during the 19th century played a significant role in initial introduction of the bacterium to several regions worldwide, such as in Madagascar where it persists until today[15,16,52,53]. As such, the possibility of maritime introductions of plague into London and Marseille during the second pandemic vastly expands the breadth of potential geographic source(s) for these strains. Nevertheless, the phylogenetic positioning of the BED and OBS genomes within the "post-BD lineage" and in relation to other second pandemic isolates suggests they arose within Europe or its vicinity.

We identified a 49-kb deletion within both BED and OBS genomes (Fig. 4b), which caused the loss of two virulence-associated genes, *mgtB* and *mgtC* (Fig. 4a). This deletion could not be identified in other second pandemic or modern strains in our dataset (Supplementary Fig. 21). The inferred virulence potential of *mgtB* and *mgtC* genes is associated with intracellular survival of *Y. pestis* within macrophages[40,54]. Their co-expression has been shown to affect the virulence exerted by other pathogenic enterobacteria under laboratory conditions[55,56] and both genes have been proposed as potential drug targets[40,57]. Moreover, the function of *mgtB* was shown to be temperature-dependent, being active at 37 °C but not at 20 °C[58], suggesting its loss affects the bacterium in warm-blooded hosts. Intriguingly, a 45-kb deletion in the same region was identified in genomes associated with late outbreaks of the first plague pandemic (6th–8th century AD)[28], which sets it as a candidate for convergent evolution and raises questions regarding its functional importance. Given that all genomes displaying this deletion were obtained from plague victims, including the Great Plague of Marseille (1720–1722 AD) that is known to have caused high mortality, its occurrence may not have reduced the pathogen's virulence, particularly since genome decay is a well-established characteristic of *Y. pestis* evolution[59,60]. Nevertheless, since both lineages that show this deletion are likely extinct, its functional characterisation will be of importance to evaluate potential effects on maintenance in mammalian and arthropod hosts, in Europe, during the first and second pandemics.

The second plague pandemic has arguably caused the highest levels of mortality of the three recorded plague pandemics[1,61]. It serves as a classic historical example of rapid infectious disease emergence, long-term local persistence and eventual extinction

for reasons that are currently not understood. We have shown that extensive sampling of ancient *Y. pestis* genomic data can provide direct molecular evidence on the genetic relationships of strains present in Europe during that time. In addition, we provide relevant information regarding the initiation and progression of the second pandemic and suggest that a single source reservoir may be insufficient to explain the breadth of epidemics and *Y. pestis*' genetic diversity in Europe during the 400-year course of the pandemic. Although certain key regions in western Eurasia remain under-sampled for ancient *Y. pestis* DNA, namely the eastern Mediterranean, Scandinavia and the Baltics, vast amounts of high-quality genomic data are becoming increasingly available. Their integration into disease modelling efforts, which consider vector transmission dynamics[62,63], climatic[12,64,65] and epidemiological data[66], as well as a critical re-evaluation of historical records[67], will become increasingly important for better understanding the second plague pandemic.

## Methods

**Tooth sampling, DNA extraction and *Y. pestis* qPCR screening.** Laboratory work was primarily performed in the dedicated aDNA facilities of the Max Planck Institute for the Science of Human History in Jena. Part of the sampling and DNA extractions were performed at aDNA facilities of the ArchaeoBioCenter of the Ludwig Maximilian University of Munich and aDNA facilities of the University of Cambridge, Department of Archaeology.

One-hundred and eighty teeth from nine sites located in England (BED), France (TRP), Germany (NAB, MAN, STA, LBG, BRA), Russia (LAI) and Switzerland (STN) spanning the 14th–17th centuries (see Supplementary Note 1) were sectioned in the cementoenamel junction, and 30–70 mg of powder was removed from the surface of the pulp chamber using a dentist drill. This powder was then used for DNA extraction, using a protocol optimised for the retrieval of short fragments that are characteristic of ancient DNA[68]. Tooth powder was incubated in 1 ml of lysis buffer (0.45 M EDTA, pH 8.0, and 0.25 mg/ml proteinase K) overnight (12–16 h) at 37 °C. Then, DNA was bound to the silica membrane of spin columns using 10 ml of GuHCl-based binding buffer as described before[68], followed by a purification that was performed using either the MinElute purification kit (Qiagen) or the High Pure Viral Nucleic Acid Large Volume Kit (Roche). DNA was eluted in 100 μl of TET (10 mM Tris-HCl, 1 mM EDTA pH 8.0, 0.05% Tween 20). Extraction blanks and a positive extraction control (cave bear specimen) were taken along for every extraction batch. All extracts were then evaluated for PCR inhibition, by spiking 2 μl of each extract in a qPCR reaction containing a standard of known concentration[17]. None of the extracts showed signs of PCR inhibitions and, therefore, all were tested by qPCR for the presence of the plasminogen activator gene (*pla*), located on the *Y. pestis*-specific pPCP1 plasmid using a published protocol[17]. PCR products were not sequenced as all putatively positive samples were subsequently evaluated through whole-genome enrichment and next-generation sequencing. All extraction and PCR blanks were free of amplification products.

In addition, 26 specimens from the Augustinian Friary in Cambridge (NMS) were sampled and DNA was extracted at the University of Cambridge. Roots were sawed from teeth using a sterile dremel cutting wheel and a UV-irradiated toothbrush was then used to briefly brush the roots with 5% w/v NaOCl. Subsequently, roots were soaked in 6% w/v bleach for 5 min, then rinsed twice with ddH₂O, and finally soaked in 70% ethanol for 2 min. The roots were then transferred to a sterile paper towel and UV irradiated for 50 min on each side. After irradiation, teeth were weighed and subsequently transferred in 5-ml or 15-ml tubes for DNA extraction. DNA extraction was carried out as follows: 2 ml of EDTA (0.5 M, pH 8.0) and 50 μl of Proteinase K (10 mg/ml) were used for every 100 mg of sample. Extractions were then incubated at room temperature for 72 h. Extracted DNA was concentrated using the Amicon Ultra-15 concentrators with a 30-kDa filter, down to 250 μl. DNA was then purified using the MinElute PCR purification kit (Qiagen) according to manufacturer's instruction. For the elution step, column-bound DNA was incubated with 100 μl of Elution buffer for 10 min at 37 °C.

**Non-UDG library preparation and metagenomic screening with HOPS.** The following protocol was carried out in the ancient DNA facility of the University of Cambridge, Department of Archaeology.

Non-UDG libraries were prepared for the NMS samples (Augustinian Friary, Cambridge; Supplementary Table 2) with the NEBNext® Library Preparation Kit for 454 (E6070S, New England Biolabs, Ipswich, MA) using a modified version of the manufacturer's protocol[69]. Adaptors were constructed as previously described[21]. Indexing PCR reactions were set up as follows: 50 μl of DNA library, 1× PCR buffer, 2.5 mM MgCl₂, 1 mg/ml BSA, 0.2 μM in PE 1.0, 0.2 mM dNTP each, 0.1 U/μl HGS Taq Diamond and 0.2 μM indexing primer, with the following cycling conditions: 5 min at 94 °C, followed by 18 cycles of 30 s each at 94 °C, 60 °C

and 68 °C, with a final extension of 7 min at 72 °C. Amplified products were purified using the MinElute kit (Qiagen) and DNA was eluted in 35 µl EB. The indexed libraries were then quantified using the Quant-iT™ PicoGreen® dsDNA kit (P7589, Invitrogen™ Life Technologies) on the Synergy™ HT Multi-Mode Microplate Reader with Gen5™ software. Subsequent shotgun sequencing of these libraries was carried out on an Illumina NextSeq500 platform (using the High-Output kit 1 × 75 cycle chemistry) at the University of Cambridge Biochemistry DNA Sequencing Facility.

The program MALT (version 0.4.0)[18], integrated in the pathogen screening pipeline HOPS[19], was used to assess the presence of *Y. pestis* DNA in the NMS specimens. A custom NCBI RefSeq (November 2017) database was used for running MALT, including all bacterial and viral assemblies marked as complete, a selection of eukaryotic pathogen genomes, as well as the human reference sequence (GRCh38). Genomes with keywords such as "unknown" were removed. A total of 15,361 genomes were retained in the database. Pre-processed shotgun NGS reads (.fastq) were used as input and the parameters were set as follows: 85 for the minimum percentage identity (-minPercentIdentity), 1 for the minimum support (-minSupport), using a top percentage value of 1 (-topPercent), a semi-global alignment mode, and with all remaining parameters set to default. The resulting ".rma6" output files were automatically post-processed with MALTExtract (in HOPS) against a list of 100 target bacterial pathogen species, and the resulting profiles were qualitatively assessed within HOPS for the number of aligning reads, the read edit distance against different taxa and the presence of aDNA damage patterns[19].

**UDG library preparation and *Y. pestis* whole-genome capture.** All putative *Y. pestis*-positive samples were subsequently converted into Illumina double-stranded DNA libraries as described before[21], using a starting volume of 50–60 µl, with an initial USER (New England Biolabs) treatment step, where UDG was used in combination with endonuclease VIII to excise uracil nucleotides that result from post-mortem DNA damage[20,70]. Subsequently, full UDG-treated and partially UDG-treated libraries were quantified on a qPCR using the IS7/IS8 primer combination. Following, a double-indexing step was performed where libraries were split into multiple PCR reactions based on their initial quantification[71], in order to ensure maximal amplification efficiency. Every reaction was assigned a maximum input of 2 × 10^10 DNA molecules. A unique index combination (index primer containing a unique 8-bp identifier) was assigned to every library, and a 10-cycle amplification reaction was used to attach index combinations to DNA library molecules using *Pfu Turbo Cx Hotstart DNA Polymerase* (Agilent). PCR products were purified using the MinElute DNA purification kit (Qiagen), and eluted in TET (10 mM Tris-HCl, 1 mM EDTA pH 8.0, 0.05% Tween 20). After indexing, all libraries were amplified using *Herculase II Fusion DNA Polymerase* (Agilent) to a concentration of 200–300 ng/µl, in order to achieve 1–2 µg of DNA in a total of 7 µl. Products were again purified using the MinElute DNA purification kit (Qiagen), and eluted in TET (10 mM Tris-HCl, 1 mM EDTA pH 8.0, 0.05% Tween 20). In-solution whole-genome *Y. pestis* capture was then performed as described previously[22], where the following genomes were used as templates for probe design: CO92 chromosome (NC_003143.1), CO92 plasmid pMT1 (NC_003134.1), CO92 plasmid pCD1 (NC_003131.1), KIM10 chromosome (NC_004088.1), Pestoides F chromosome (NC_009381.1) and *Y. pseudotuberculosis* IP 32953 chromosome (NC_006155.1). DNA captures were carried out on 96-well plates. Each sample was either captured in its individual well, or pooled with maximum one more sample from the same site. Capture enrichment was carried out for two rounds, except for sample NMS002 that was captured for one round. Blanks with non-overlapping index combinations were captured together.

**Sequencing and read processing.** After capture, all products were sequenced on an Illumina NextSeq500 platform using (1 × 151 + 8 + 8 cycles or 1 × 76 + 8 + 8 cycles) or on a HiSeq4000 (using 1 × 76 + 8 + 8 cycles or 2 × 76 + 8 + 8 cycles). Preprocessing of de-multiplexed reads was performed on the automated pipeline EAGER (v1.92.55)[72] and involved the removal of Illumina adaptors and read merging using AdapterRemoval v2 (ref. [73]), as well as filtering all reads for sequencing quality (minimum base quality of 20) and length (to retrieve only reads ≥30 bp). Subsequently, reads were mapped with BWA (version 0.7.12)[74], implemented in EAGER, against the CO92 reference genome (NC_003143.1)[3] using stringent parameters (-n 0.1, -l 32) for genome reconstruction and mean coverage calculation and more lenient parameters (-n 0.01, -l 32) for inspection of regions surrounding potential variants. Reads with mapping quality lower than 37 (-q) were removed using SAMtools (http://samtools.sourceforge.net/), and PCR duplicates were removed using the MarkDuplicates tool (http://broadinstitute.github.io/picard/). Prior to SNP identification, raw pre-processed reads from partially-UDG-treated libraries were trimmed for 2-bp at both ends to remove sites that could be affected by aDNA damage and, subsequently, were re-filtered for length and re-mapped using stringent parameters.

**SNP calling and phylogenetic analysis.** SNP calling was performed using the UnifiedGenotyper of the Genome Analysis Toolkit (GATK v3.5)[75]. Our newly reconstructed genomes were analysed alongside previously published *Y. pestis* genomes, which included a modern-day dataset of 233 genomes[23–27,48] (as listed in

ref. [28]), three Bronze Age strains[31], a 2nd- to 3rd-century AD isolate from the Tian Shan mountains in Kyrgyzstan[30], one Justinianic strain (Altenerding 2148)[29], 15 previously published historical genomes from the second plague pandemic[8–10,14] and a *Y. pseudotuberculosis* strain (IP32953)[60] that was used as outgroup for rooting the phylogeny. A vcf file was produced for every genome using the "EMIT_ALL_SITES" option, which generated a call for every position present in the reference genome. Furthermore, we used the custom java tool MultiVCFAnalyzer v0.85 (ref. [33]) (https://github.com/alexherbig/MultiVCFAnalyzer) to produce a SNP table of variant positions across all genomes analysed, using the following parameters: for homozygous alleles, a SNP would be called when covered at least 3-fold with a minimum genotyping quality of 30, and for heterozygous alleles, a variant would be called when 90% of reads would support it. In cases where none of the parameters would be met, an "N" would be inserted in the respective genomic position. In addition, we omitted previously defined noncore regions, as well as annotated repetitive elements, homoplasies, tRNAs, rRNAs and tmRNAs from our SNP analysis[16,23]. In the present dataset, a total of 7,510 variant positions were identified. Subsequently, the annotation as well as the effect of each SNP was determined through the program SnpEff v3.1i (ref. [76]).

We used a SNP alignment produced by MultiVCFAnalyzer v0.85 to construct phylogenetic trees using the ML and maximum parsimony (MP) methods. Up to 3% missing data were included in the analysis (97% partial deletion), resulting in a total number of 6,058 SNPs used for phylogenetic reconstruction. The MP phylogeny was produced in MEGA7 (ref. [77]) in order to make a first assessment of genome topologies. The ML phylogenies were constructed with the program RAxML (version 8.2.9)[78] using the Generalised Time Reversible (GTR)[79] model with four gamma rate categories and 1000 bootstrap replicates to assess tree topology support.

**Reanalysis of recently published non-UDG *Y. pestis* genomes.** A recent study described the phylogenetic positioning and SNP analysis of five 14th century *Y. pestis* genomes[8]. As these genomes were non-UDG treated, they were reanalysed here using different criteria compared to other second pandemic and modern genomes in our dataset. Read pre-processing and merging was done as described in the above section "Sequencing and read processing". In addition, read mapping against the CO92 reference genome (NC_003143.1) was performed using more lenient parameters in BWA[80] (-n 0.01, -l 16) than the ones previously reported[8], to account for ancient DNA deamination within mapping reads. In our view, the usage of strict BWA mapping parameters for non-UDG data (i.e. −n 0.1) could potentially introduce a reference bias to the analysis, which could in turn affect SNP discovery and phylogenetic inferences. PCR duplicates were removed from all five datasets using MarkDuplicates (http://broadinstitute.github.io/picard/) and reads were filtered for mapping quality (q 37) using SAMtools (http://samtools.sourceforge.net/). The obtained mean coverage for each of the five re-analysed genomes was: 3.4-fold for BSS31 (27.8% covered 5-fold), 6.7-fold for SLC1006 (59.1% covered 5-fold), 30.5-fold for OSL-1 (91.7% covered 5-fold), 38.1-fold for Ber37 (95.2% covered 5-fold) and 46.1-fold for Ber45 (94.1% covered 5-fold). In addition, the obtained average fragment lengths for the five re-analysed genomes were as follows: 52.2 bp for BSS31, 71.5 bp for SLC1006, 108 bp for OSL-1, 61.9 bp for Ber37 and 69.7 bp for Ber45. Before SNP calling, MapDamage2.0 (ref. [81]) was used to rescale base qualities, primarily on the extremities of mapped reads, to account for sites that could potentially be affected by aDNA deamination. Subsequently, SNPs were called using GATK and the resulting vcf files were comparatively assessed in MultiVCFAnalyzer v0.85 (ref. [33]) to compile a SNP table including all genomes in the dataset as described in the above section "SNP calling and phylogenetic analysis".

**Qualitative SNP assessment in UDG-treated data using SNPEvaluation.** A frequent challenge faced when using ancient metagenomic datasets to reconstruct bacterial genomes is a strong environmental signal that can interfere with SNP assignments, especially in low-coverage data[29]. Such an effect can interfere with phylogenetic analyses by creating artificial branch lengths, which can in turn affect evolutionary inferences. As such, in order to avoid erroneous SNP assignments, we qualitatively evaluated all private SNP calls for the newly reconstructed genomes that were used for phylogenetic analysis in this study (minimum 50% of the genome covered 5-fold (Table 1)). We used the recently developed SNPEvaluation tool (https://github.com/andreasKroepelin/SNP_Evaluation) to compare the SNP profiles that arise for each dataset under both stringent (BWA parameters -n 0.1, -l 32) and more lenient (BWA parameters: -n 0.1, -l 16) mapping parameters. Subsequently, the region around each SNP was evaluated within a 50-bp window, and was accepted as true when fulfilling the following criteria: (i) the ratio of coverage between the lenient and stringent mapping was not higher than 1.00, (ii) there were no heterozygous positions within this window when considering a high stringency mapping and (iii) no missing regions/bases were observed within close proximity to the identified SNP (see Supplementary Data 2). Note that the above criteria in SNPEvaluation have been determined and evaluated in UDG-treated metagenomic data[28] and, therefore, would need to be re-adapted for non-UDG-treated data that are heavily affected by aDNA deamination.

**Heterozygosity estimates**. Heterozygous variant calls were investigated given the disparity of branch lengths observed in certain newly reconstructed and previously published genomes (see Supplementary Figs. 14 and 16). Our approach takes into account the "haploid" nature of prokaryotic genomes, suggesting that "heterozygous" SNPs could either arise as a result of mixed infections or from erroneous mapping of DNA reads that belong to closely related bacterial contaminants. We performed SNP calling with the UnifiedGenotyper in GATK[75], using the "EMIT_ALL_SITES" option that generated a call for all positions in the reference genome. We then used MultiVCFAnalyzer v0.85 (ref. [33]) to compile a SNP table of variant positions with allele frequencies 10–90% across our dataset, hence accounting for all ambiguous heterozygous positions. Histograms of allele frequencies for all SNPs with <100% read support were constructed with R v3.4.1 (ref. [82]) using representative genomes from all sites.

**Estimates of substitution rate variation in Y. pestis**. In order to calculate the substitution rate variation across Y. pestis isolates associated with the second pandemic, we first assessed the temporal signal across Branch 1 that includes all genomes from both the second and third plague pandemics. For this, we computed an ML phylogeny in RaxML[78] using all Branch 1 genomes[3,8–10,14,16,23,48,83,84] (modern + ancient n = 79), with the exception of the BD genomes TRP002 and OSL-1 that showed possible environmental contamination to be affecting their SNP calls. In addition, we used the strain 2.MED KIM10 (Branch 2) as outgroup for rooting the tree. Variant positions across this set of genomes were used for the analysis, allowing for up to 3% missing data (550 SNPs). We used TempEst v1.5 (http://tree.bio.ed.ac.uk/software/tempest/) for calculating the root-to-tip regression in relation to specimen or sampling ages. The calculated correlation coefficient (R) and $R^2$ values were 0.57 and 0.33, respectively, which permitted the proceeding with molecular dating analysis.

The Bayesian framework BEASTv1.8 (ref. [85]) was used to assess the substitution rate variation across the Y. pestis Branch 1 using the 2.MED KIM10 as outgroup. Our BEAUti setup took into consideration all archaeological, radiocarbon and sampling dates of both ancient and modern genomes (Supplementary Data 6) that were used as calibration points for the Bayesian phylogeny. Divergence dates for each node in the tree were estimated as years before the present, where the year 2005 was set as the present since it represents the most recently isolated modern Y. pestis strain on Branch 1 (1.ORI MG05). Monophyletic clades were defined based on the ML phylogeny (Supplementary Fig. 12). The GTR[79] substitution model (4 gamma rate categories) and a lognormal relaxed clock (clock rate tested and strict clock rejected in MEGA7[77]) were used to set up two separate analyses using the coalescent constant size[86] and coalescent Bayesian skyline[87] demographic models. For each analysis, three independent chains of 50,000,000 states each were carried out and then combined using LogCombiner to ensure run convergence, with 10% burn-in. In addition, we estimated marginal likelihoods to determine the best demographic model for our dataset using path sampling and stepping stone sampling (PS/SS) implemented in BEAST v1.8 (ref. [85]). For this, each of the described runs was carried out for an additional 50,000,000 states (500,000 states divided into 100 steps) using an alpha parameter of 0.3, which determined the coalescent Bayesian skyline model as better fit for the current dataset. The results produced by the run considering this demographic model were then viewed in Tracer v1.6 (http://tree.bio.ed.ac.uk/software/tracer/) to ensure all relevant effective sample sizes (ESS) were >200. We used TreeAnnotator[85], to produce a maximum clade credibility (MCC) phylogeny using the best-fit model with 10% burn-in, which resulted in the processing of 13,503 trees. The MCC tree was viewed and modified in FigTree v1.4 (http://tree.bio.ed.ac.uk/software/figtree/) where branch lengths were represented as a function of age and mean rates were used to colour individual branches. Finally, the skyline plot was produced and viewed using Tracer v1.6 (http://tree.bio.ed.ac.uk/software/tracer/) after resampling states at a lower frequency (every 100,000) using LogCombiner[85].

**Gene presence/absence and deletion analysis**. In order to investigate the virulence-associated gene profiles of the newly reconstructed second pandemic genomes, the highest quality (coverage) genome from every site (LAI009, NAB003, TRP002, MAN008, STA001, NMS002, LBG002, STN014, BRA001, BED030) was used for comparison with each other and with previously published representatives of ancient (London BD 8124/8291/11972, OSL-1 Ber45, London 6330, Bolgar 2370, Barcelona 3031, Ellwangen 549_O, OBS137, RISE509, RT5, Altenerding 2148) and modern (1.ORI-CO92, 0.PE2-PESTF, 0.PE4-Microtus 91001) Y. pestis isolates as well as a Y. pseudotuberculosis strain (IP32953). All listed genomes were re-mapped against the CO92 chromosomal reference genome (NC_003143.1) without the use of a mapping quality filter of (-q 0). The coverage across 80 chromosomal and 43 plasmid virulence-associated and evolutionary determinant genes that were previously defined[36] was calculated using bedtools[88]. The results are plotted in the form of a heatmap using the ggplot2 (ref. [89]) package in R version 3.4.1 (ref. [82]) and can be viewed in Fig. 4. In addition, we used BWA-MEM[80] to explore the precise coordinates of observed gene or region losses in all affected genomes using default parameters. For the visualisation of an identified deletion across BED and OBS isolates, we computed the average coverage across 3,000-bp windows in representative Y. pestis genomes from all analysed periods of the second pandemic, and subsequently used the program Circos[90] to produce coverage plots of a 20-fold maximum coverage. The coverage plots were arranged in chronological order as follows: LAI009, London BD 8124/8291/11972, Ber45, Bolgar 2370, MAN008, STA001, NMS002, ELW098, LBG002, STN014, BRA001, BED030, OBS137 and the reference genome CO92.

**Reporting summary**. Further information on research design is available in the Nature Research Reporting Summary linked to this article.

## Data availability

Raw sequencing data of the deep-sequenced genomes are available on the European Nucleotide Archive under project accession number PRJEB29990 . Other data supporting the findings of the study are available in this article and its Supplementary Information files, or from the corresponding authors upon request.

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

## Acknowledgements

We thank Aditya K. Lankapalli and Stephen Clayton for computational analysis support. We thank Guido Brandt, Antje Wissgott, Cäcilia Freund and Marta Burri for laboratory support. We are grateful to Monica Green for critical comments on the manuscript. We thank Hans Sell and Michelle O'Reilly for graphics support. We thank Rafail' M. Fattahov for facilitating excavations of the Laishevo III archaeological site, Ayrat Sitdikov for providing access to the Laishevo III skeletal assemblage and Elizaveta V. Volkova for assisting with sampling of skeletal material. In addition, we would like to thank Joke Somers for the anthropological analysis and sampling of the Stans individuals. We thank Bettina Jungklaus for providing the samples from Brandenburg an der Havel, Bernd Trautmann for morphological analyses, Jochen Haberstroh and Mathias Hensch for providing archaeological information, and the staff of the SAPM for support during sample collection. We also thank Benoît Kirschenbilder, for his initial involvement in this project in association with the Toulouse archaeological site (16 rue des Trente Six Ponts). The fieldwork at the New Churchyard was led by Alison Telfer, and radiocarbon dating was carried out by 14CHRONO Centre, The Queen's University, Belfast, Northern Ireland. Analysis of radiocarbon dates from New Churchyard was performed by Derek Hamilton of the Scottish Universities Environmental Research Centre (SUERC), East Kilbride, Scotland, and Peter Marshall of Historic England. Radiocarbon dating for the Stans collection was performed at the LARA laboratory of the Department of Chemistry and Biochemistry at the University of Bern. Radiocarbon dating for all other material was performed in the Curt-Engelhorn-Zentrum Archäometrie gGmbH in Mannheim, Germany. The Cambridge work is supported by the Wellcome Trust (Award no. 2000368/Z/15/Z) and St. John's College, Cambridge (J.E.R., T.K., C.C., C.L.S.); the European Union through the European Regional Development Fund (Project No. 2014-2020.4.01.16-0030) (C.L.S.); and the Estonian Research Council personal research grant (PRG243) (C.L.S). M.A.S., M.K., K.I.B. and J.K. were supported by the Max Planck Society and the ERC starting grant APGREID (to J.K.). R.T., A.H. and K.I.B. were supported by the Max Planck Society.

## Author contributions

M.A.S., M.K., R.I.T., M.Ha., K.I.B. and J.K. designed the study. M.A.S., M.K., R.T., E.A.N., C.L.S., G.U.N. and P.M.-A. performed laboratory work. M.A.S., M.K., A.A.V., F.M.K. and A.H. performed data analysis. D.W., A.A., N.C., H.F., M.G., R.H., M.He., K.v.H., S.A.I., S.K., E.L.K., J.P., J.E.R., D.C., S.L. and M.Ha. performed anthropological analysis, as well as identified and provided access to appropriate archaeological material. A.A., J.S., K.v.H., C.L. and C.C. facilitated excavations and provided access to unpublished archaeological information. T.K., M.Ha., A.H., K.I.B. and J.K. supervised different aspects of the study. M.S., M.K. and K.I.B. wrote the paper with contribution from all co-authors.

## Additional information

**Competing interests:** The authors declare no competing interests.

