## [Peer Review File · Nature Communications]

Reviewers' comments:

Reviewer #1 (Remarks to the Author):

In this study, Spyrou and colleagues present the largest genome collection of second-pandemic plague strains, including 32 original genome sequences with coverage ranging 1.1-80.1-fold. Together with 12 second-pandemic genomes previously characterized, this helps address long-lasting controversies regarding single vs multiple entries of the plague into Europe and the strains phylogeography. The second-pandemic plague strains have haunted our populations during the late Middle Ages until the 18th century, and were responsible for millions of deaths. There is no doubt that the findings of Spyrou and colleagues will be of significant interest for a broad readership. In addition to leverage state-of-the-art molecular methods in ancient DNA research, including post-mortem damage removal, dual library indexing and target-enrichment, the authors carry out rigorous computational analyses, scrutinizing the sequence alignments produced for the presence of potentially spurious reads. Their conclusions are sound and I find it particularly hard to not like this manuscript. I thus recommend publication. I would however suggest considering the four following points to improve the work even further.

1-it is unclear to me whether the authors included the recently published genomes from Manouchi et al. PNAS 2018. It does not seem that it is the case. Yet, these include important strains from the C14th and from countries not represented here, such as Norway, the Netherlands and Italy (plus an additional French site). Assuming that the coverage and data quality are sufficient, the authors should include these additional genomes to their own dataset so as to complete the geographical range considered and, thus, their phylogeographic study.

2-as the authors rely on target-enrichment techniques, and as the genetic diversity detected at the SNP level is fairly limited, they should discuss about other possible sources of genetic variation that are not possible to assess here (e.g. structural variation).

3-the authors provide evidence for extreme acceleration of mutation rates during the evolution of some of the second-pandemic strains. This is an extremely interesting observation, especially as bacterial hypermutability is a well-discussed process both in evolutionary biology and the host-pathogen literature. Hypermutable strains can benefit from a fitness advantage due to the quicker emergence of beneficial mutations, which could be helpful in the co-evolutionary arm race against their host. At the same time, mutators will eventually be affected by the emergence of deleterious variants hitch-hiked on the beneficial genomic background, possibly counter-balancing the fitness effects of the beneficial allele. This can ultimately be avoided by mutations reversing the mutator phenotype. The authors should speculate whether such mechanisms could be at play here, so as to explain the cycles of acquisition/loss of hypermutability detected along independent tree branches. If so, some of the variants emerging following episodes of hypermutability are likely to have been beneficial and should be non-synonymous. Is there any mutation amongst those described that fit this scenario? The authors should also speculate about the molecular mechanisms underlying the emergence of mutators (eg are genes involved in DNA repair mechanisms affected?). Finally, the convergent loss of the 49kb genomic block is interesting in the light of the hypermutability phenotype. Is that genomic region particularly prone to instability? is the sequence composition just on par with the genomic average?

4- the authors should quantitatively assess the impact of the probe design on the efficacy of target-enrichment. This could be easily done by calculating the edit distance between *Y. pestis* (the target) and *Y. pseudotuberculosis* (used while designing probes) and test whether the local depth-of-coverage decreases as the distance increases (perhaps using %GC as a covariate). Finding no such trend would plead against any significant reference bias.

5- I am curious to see the demographic skyline profile reconstructed as part of the rate analysis. Would be interesting to comment at the likely episodes of bursts and collapses of the plague during history, in the face of the known demography of the human reservoir.

Reviewer #2 (Remarks to the Author):

Spyrou and colleagues produced an impressive dataset of ancient genomes from the second pandemic plague pandemic. The analyses are basic and the paper is primarily descriptive. To me, the key results are the identification of the oldest (and basal) second-pandemic isolate in Eastern Europe, the low genetic diversity during the early stages of the pandemic with three Black Death identical genomes from different parts of Europe, and the confirmation that the molecular clock of *Y. pestis* undergoes dramatic accelerations.

Additionally, the authors identify a genomic deletion of the MgtCB operon in two late 2nd pandemic strains, which is very similar to one they previously identified in a late first-pandemic isolate. They interpret this deletion as likely evidence for convergent evolution towards lower virulence during the later stages of both pandemics.

I do not have any particular problem with the data and the analyses presented. The paper represents an interesting contribution to our understanding of the second pandemic, and the data should prove precious for future investigations of the epidemiology and evolution of plague. Below, please find a couple of minor concerns.

Convergent evolution: I appreciate that the observation of a similar genomic deletion of the MgtCB operon both in late strains from the first and second pandemic is interesting. Though, this could also be a coincidence. *Y. pestis* has many 'virulence factors' and it is not obvious to me why reduction in virulence should happen through deletion of the exact same genes. In the absence of any supportive epidemiological information, the claim in the abstract about convergent evolution for reduced virulence feels too strongly worded to me.

Virulence factors: I was wondering how the list of virulence factors was compiled. Is this simply the list provided by Zhou & Yang 2009 (reference 33 in the manuscript), or has additional work gone into this. I am asking because the Zhou and Yang review is now ten years old and uses a fairly loose definition for 'virulence factors'.

We thank both reviewers for their valuable comments that have ultimately improved the quality of our paper. Please find our point-by-point response to individual queries below. In addition, all newly added or altered parts of the main text and supplementary information sections are marked in yellow.

In our latest version of our paper we have integrated the newly published genomes by Namouchi et al¹, and our re-analysis of their data is now described in a dedicated methods section named “Re-analysis of recently published non-UDG *Y. pestis* genomes (14th century)”. Moreover, we have included newly generated genomes from an additional site (Augustinian Friary, Cambridge), which now increases the total number of sites we investigated to ten, and the number of new genomes we report here to 34.

Reviewers' comments:

Reviewer #1 (Remarks to the Author):

I-it's unclear to me whether the authors included the recently published genomes from Manouchi et al. PNAS 2018. It does not seem that it is the case. Yet, these include important strains from the C14th and from countries not represented here, such as Norway, the Netherlands and Italy (plus an additional French site). Assuming that the coverage and data quality are sufficient, the authors should include these additional genomes to their own dataset so as to complete the geographical range considered and, thus, their phylogeographic study.

At the time of our first submission, the data from Namouchi et al. were too recent for integration into our analyses. However, we have since performed a reanalysis of these new data, and attempted their integration in our study. We have added a new methods section that describes our re-analysis, and the criteria we apply for quantitating genetic differences (see “Re-analysis of recently published non-UDG *Y. pestis* genomes (14th century)” Lines 532 - 554). Notably, our methods differ from those of Namouchi et al., and this led to some differences in the trees presented in both papers. Specifically, we find that two of five genomes described (BSS31 from Italy and SLC1006 from France) suffer from a high amount of heterozygosity, which suggests a degree of contamination from closely related environmental bacteria (Supplementary figure 5). Such an effect has been previously described in ancient metagenomic datasets^{2,3}. This contamination is further evidenced by the atypically long branch lengths that these genomes show compared to other second pandemic and modern *Y. pestis* genomes sequenced to date (Supplementary figures 4 & 6). In addition, these very phenomena also formed the basis for the exclusion of four newly reconstructed genomes from further analysis in our study (NAM005, BRA003, SNT004 and STN011).

Of note, Namouchi et al do not describe similarly problematic branch lengths in their study. They employed a technique of SNP exclusion, where SNPs are omitted when they are identified in proximity of ≤ 10 bp from each other, without any further assessment. This approach could potentially be problematic since variant exclusion without qualitative assessment could lead to the omission of true variants as well as potential misidentification of false-positive diversity. Instead our results here demonstrate the importance of a method where SNPs that cluster in “problematic”

regions are qualitatively assessed and regarded as potential “false-positives” or “true-positives” instead of being immediately omitted from a dataset.

The most significant discrepancy in the data relates to the presence of genetic diversity during the Black Death that Namouchi et al present based on two unique SNPs identified in a low-coverage (3-fold based on our re-analysis) genome from mid-14th-century Italy. Here, we paid special attention and qualitatively assessed these two variants given the implications of this result for *Y. pestis* microdiversity during the Black Death (Supplementary figure 8). Through BLASTn analysis of reads overlapping those regions we could show that both have 100% identity to environmental or other enteric bacteria, but not to *Y. pestis* (Supplementary data 4 & 5). This result did not surprise us given the contaminant signal revealed through our reanalysis described above. On this basis we argue that the diversity reported by Namouchi et al is not supported (lines 189 – 195).

2-as the authors rely on target-enrichment techniques, and as the genetic diversity detected at the SNP level is fairly limited, they should discuss about other possible sources of genetic variation that are not possible to assess here (e.g. structural variation).

We agree with this point. We have now included a sentence within our results section to acknowledge other sources of genetic variation that were not accounted for in the present study. The new sentence is as follows:

“In addition, we note that structural rearrangements could provide alternative means of genetic diversity. Although differences in genome architecture are vastly abundant among modern *Y. pestis* genomes, their assessment in ancient *Y. pestis* is limited by the short read aDNA data produced here”, (Lines 195-199).

3-the authors provide evidence for extreme acceleration of mutation rates during the evolution of some of the second-pandemic strains. This is an extremely interesting observation, especially as bacterial hypermutability is a well-discussed process both in evolutionary biology and the host-pathogen literature. Hypermutable strains can benefit from a fitness advantage due to the quicker emergence of beneficial mutations, which could be helpful in the co-evolutionary arm race against their host. At the same time, mutators will eventually be affected by the emergence of deleterious variants hitch-hiked on the beneficial genomic background, possibly counter-balancing the fitness effects of the beneficial allele. This can ultimately be avoided by mutations reversing the mutator phenotype. The authors should speculate whether such mechanisms could be at play here, so as to explain the cycles of acquisition/loss of hypermutability detected along independent tree branches. If so, some of the variants emerging following episodes of hypermutability are likely to have been beneficial and should be non-synonymous. Is there any mutation amongst those described that fit this scenario? The authors should also speculate about the molecular mechanisms underlying the emergence of mutators (eg are genes involved in DNA repair mechanisms affected?). Finally, the convergent loss of the 49kb genomic block is interesting in the light of the hypermutability phenotype. Is that genomic region particularly prone to instability? is the sequence composition just on par with the genomic average?

We thank the review for these suggestions. Apart from assessing the substitution rate variation across Branch 1 of the *Y. pestis* phylogeny, our original submission also reports a SNP effect analysis to characterise the mutations occurring in those internal branches. The results of this analysis can be viewed in Supplementary data 7-9. However, comparisons of synonymous vs non-synonymous SNP proportions across the phylogeny, which essentially encompasses a dNdS analysis, goes beyond the analytical scope of this manuscript. The first large-scale genomic study of *Y. pestis*⁴ reported high disparities between branch lengths in different areas of the tree, which was interpreted as revealing high mutation rate variation as a global phenomenon for *Y. pestis* lineages. This variation was not attributed to the presence of few hypermutable strains, or to evidence for natural selection in certain tree branches, but rather to increased replication cycles experienced during epidemic or pandemic events. Hence we have structured our arguments within this framework. We acknowledge that more in-depth analyses of the patterns of rate variation deserve a detailed re-evaluation, especially in light of increasing genomic data from historical pandemic events. We are currently conducting a separate study on comparative genomics across all sequenced *Y. pestis*, and this may yield further details related to this phenomenon.

Considering our three identified hypermutable branches on their own, we do not find an overrepresentation of non-synonymous compared to synonymous SNPs (See Supplementary data 7–9). When assessing these SNP effects, we do identify non-synonymous substitutions that are potentially associated with DNA replication and repair mechanisms (such as in genes *recC*, *recF* and *uvrB*); however, we would prefer not to speculate further on their functional significance for two reasons: (1) The functional outcome of these SNPs is unknown, and (2) we have not investigated the occurrence of mutations in these or related genes in other branches of tree.

In addition, we thank the reviewer for suggesting an evaluation of the GC content across the 49 kb region we report as being absent in the 17th-18th London (BED) and Marseille (OBS) genomes. We have now added a new panel on the coverage plots of Figure 4b to display the areas of the genome where the GC content deviates from the mean by at least one standard deviation. We find that fluctuations in GC content are scattered all across the *Y. pestis* genome and, therefore, genomic instability due to GC disparity is unlikely to have influenced the loss of this region. In addition, the mean GC content within this region is 46.8%, which is on par with the genomic mean (47.6%). Instead, we were able to identify an *IS100* element immediately flanking one of its ends, which is consistent with previous evidence of genomic disruptions in *Y. pestis* through IS elements⁵.

4- the authors should quantitatively assess the impact of the probe design on the efficacy of target-enrichment. This could be easily done by calculating the edit distance between *Y. pestis* (the target) and *Y. pseudotuberculosis* (used while designing probes) and test whether the local depth-of-coverage decreases as the distance increases (perhaps using %GC as a covariate). Finding no such trend would plead against any significant reference bias.

Although previous studies by our group have used a *Y. pseudotuberculosis*-based probe design for array-based capture, our current paper describes a different approach. Specifically, probes were designed using a panel of modern *Y. pestis* and *Y.*

pseudotuberculosis genomes (CO92 chromosome (NC_003143.1), CO92 plasmid pMT1 (NC_003134.1), CO92 plasmid pCD1 (NC_003131.1), KIM 10 chromosome (NC_004088.1), Pestoides F chromosome (NC_009381.1) and *Y. pseudotuberculosis* IP 32953 chromosome (NC_006155.1)), in an attempt to capture an increased amount of diversity present within the *Y. pseudotuberculosis* complex. Therefore, we believe that a quantitative assessment of *Y. pseudotuberculosis* probe design would fall outside the scope of this study.

5- I am curious to see the demographic skyline profile reconstructed as part of the rate analysis. Would be interesting to comment at the likely episodes of bursts and collapses of the plague during history, in the face of the known demography of the human reservoir.

We thank the reviewer for this comment. We have generated the Bayesian coalescent skyline as suggested, which can be viewed in the figure below (now appears as Supplementary fig. 9). Note that the diversification of Branch 1 is dated to 731 yBP (95% HPD: 669 – 820 yBP). Although our skyline does show some indications of population size increase towards the period of third pandemic, we view the outcome associated with the second pandemic as a low-resolution result. We believe that our limited sample size ($n=80$) and the fact that we only use Branch 1 genomes for this analysis, has led to the production of large 95% HPD intervals for the inferred population sizes around the tree root. When such intervals are considered, the analysis could also be interpreted as consistent with a constant demographic history. We envision that future efforts to increase the number of ancient and modern *Y. pestis* data points, as well as the inclusion of additional genomic diversity from other phylogenetic branches, will offer a greater resolution.

Reviewer #2 (Remarks to the Author):

Convergent evolution: I appreciate that the observation of a similar genomic deletion of the MgtCB operon both in late strains from the first and second pandemic is interesting. Though, this could also be a coincidence. *Y. pestis* has many ‘virulence factors’ and it is not obvious to me why reduction in virulence should happen through deletion of the exact same genes. In the absence of any supportive epidemiological information, the claim in the abstract about convergent evolution for reduced virulence feels too strongly worded to me.

We thank the reviewer for this remark. We have now changed our abstract as follows: “The deletion could not be detected in extant strains, though it was identified in genomes associated with the first plague pandemic (541-750 AD), suggesting a comparable evolutionary trajectory of *Y. pestis* during both pandemic events.” (Lines 47 – 59)

We acknowledge that the deletion of a similar region between both pandemic events could have been a random phenomenon. In addition, we emphasize that most genes within the deletion carry functions that are inactive in *Y. pestis* (lines 270 – 274). However, we still consider this finding of *mgtB* and *mgtC* deletion worth mentioning since these genes seem to be functionally important for *Y. pestis* and for other enterobacteria (e.g. *S. enterica*)⁶. In addition, their loss was not observed in other ancient or modern genomes in our dataset. For increased clarity, we have produced an additional figure to visualise the presence or absence of this region among all characterised *Y. pestis* lineages (ancient + modern) that were used in this study (see Supplementary figure 11).

We would like to point out that in the discussion section of our manuscript we do consider that the *mgtB* and *mgtC* gene deletion is unlikely to have produced a decreased virulence phenotype, as in both cases the genomes showing it were isolated from epidemic contexts. Our statement reads as follows: (Lines 375 – 378). “Given that all genomes displaying this deletion were obtained from plague victims, including the Great Plague of Marseille (1720 – 1722 AD) that is known to have caused high mortality, its occurrence may not have reduced pathogen virulence particularly since genome decay is a well established characteristic of *Y. pestis* evolution.”

Instead, we speculate on its possible relationship to reservoir hosts present in Europe or the vicinity during the 17th–18th centuries AD.

Virulence factors: I was wondering how the list of virulence factors was compiled. Is this simply the list provided by Zhou & Yang 2009 (reference 33 in the manuscript), or has additional work gone into this. I am asking because the Zhou and Yang review is now ten years old and uses a fairly loose definition for ‘virulence factors’.

We thank the reviewer for this comment. The genes listed in our analysis as “virulence-associated genes” are primarily based on the list compiled by Zhou & Yang and published in 2009. We agree with the reviewer in that this list also includes genes that do not comply with a strict definition of virulence factors, but have rather been important for the evolution of *Y. pestis* and its functional differentiation from *Y.*

pseudotuberculosis. We, hence, have omitted this title from Figure 4a. The description of these genes now appears as follows (caption of Figure 4a): “Here we show an assessment of the presence or absence of 80 previously defined potential virulence and evolutionary determinants across published and newly produced *Y. pestis* genomes (chromosomal genes)”.

Such changes are also relevant in light of our newly described genome NMS002. This genome was found to lack one of the genes within this list (*inv*). The *inv* gene groups within the category of “evolutionary determinants” as it is known to be a pseudogene in all *Y. pestis*, but functionally active (and important for virulence) in *Y. pseudotuberculosis* and *Y. enterocolitica*⁷. Its loss is described in our results section as follows:

“Regarding the newly reconstructed strains, we find that most possess all analysed genes with the exception of the New Churchyard (BED) and Marseille (OBS) strains that lack the magnesium transporter genes *mgtB* and *mgtC*, as well as the Cambridge (NMS002) strain that is lacking the *inv* gene (Figure 4). While invasins are associated with epithelial colonisation of *Y. pseudotuberculosis* and *Y. enterocolitica*, it is known to have been inactivated in *Y. pestis*.” (Lines 255 – 260)

References

- 1 Namouchi, A. *et al.* Integrative approach using *Yersinia pestis* genomes to revisit the historical landscape of plague during the Medieval Period. *Proceedings of the National Academy of Sciences*, 201812865 (2018).
- 2 Vågane, A. J. *et al.* *Salmonella enterica* genomes from victims of a major sixteenth-century epidemic in Mexico. *Nat Ecol Evol* **2**, 520-528, doi:10.1038/s41559-017-0446-6 (2018).
- 3 Feldman, M. *et al.* A high-coverage *Yersinia pestis* genome from a sixth-century Justinianic plague victim. *Mol. Biol. Evol.* **33**, 2911-2923 (2016).
- 4 Cui, Y. *et al.* Historical variations in mutation rate in an epidemic pathogen, *Yersinia pestis*. *Proceedings of the National Academy of Sciences* **110**, 577-582 (2013).
- 5 Fetherston, J. D. & Perry, R. D. The pigmentation locus of *Yersinia pestis* KIM6+ is flanked by an insertion sequence and includes the structural genes for pesticin sensitivity and HMWP2. *Mol. Microbiol.* **13**, 697-708 (1994).
- 6 Groisman, E. A. *et al.* Bacterial Mg²⁺ homeostasis, transport, and virulence. *Annu. Rev. Genet.* **47**, 625-646 (2013).
- 7 Simonet, M., Riot, B., Fortineau, N. & Berche, P. Invasin production by *Yersinia pestis* is abolished by insertion of an IS200-like element within the *inv* gene. *Infect. Immun.* **64**, 375-379 (1996).

REVIEWERS' COMMENTS:

Reviewer #1 (Remarks to the Author):

The authors have satisfactorily addressed all my previous request. I recommend publication. Congrats on a nice study.

Reviewer #2 (Remarks to the Author):

This is a really nice paper. I am satisfied by the authors' responses to my queries. I also feel the authors did a good job at addressing the other reviewer's concerns.